# Sports Performance and Breathing Rate: What Is the Connection? A Narrative Review on Breathing Strategies

**DOI:** 10.3390/sports11050103

**Published:** 2023-05-10

**Authors:** Gian Mario Migliaccio, Luca Russo, Mike Maric, Johnny Padulo

**Affiliations:** 1Department of Performance, Sport Science Lab, 09131 Cagliari, Italy; ciao@migliaccio.it (G.M.M.); mike@mikemaric.com (M.M.); 2Department of Human Sciences, Università Telematica degli Studi IUL, 50122 Florence, Italy; 3Department of Biomedical Sciences for Health, Università degli Studi di Milano, 20133 Milan, Italy; johnny.padulo@unimi.it

**Keywords:** breathing pace, breathing style, slow breathing, fast breathing

## Abstract

Breathing is a natural and necessary process for humans. At the same time, the respiratory pace and frequency can vary so much, depending on the status of the subject. Specifically, in sports, breathing can have the effect of limiting performance from a physiological point of view, or, on the other hand, breathing can regulate the psychological status of the athletes. Therefore, the aim of this narrative review is to focus on the literature about the physiological and psychological aspects of breathing pace in sports performance, merging these two aspects because they are usually considered split, in order to create a new integrated vision of breathing and sports performance. Voluntary breathing can be divided into a slow or fast pace (VSB and VFB, respectively), and their effects on both the physiological and psychological parameters are very different. VSB can benefit athletes in a variety of ways, not just physically but mentally as well. It can help improve cardiovascular fitness, reduce stress and anxiety, and improve overall health and well-being, allowing athletes to maintain focus and concentration during training and competition. VFB is normal during physical training and competition, but away from training, if it is not voluntary, it can cause feelings of anxiety, panic, dizziness, and lightheadedness and trigger a stress response in the body, affecting the athlete’s quality of life. In summary, the role of breathing in the performance of athletes should be considered, although no definitive data are available. The connection between breathing and sports performance is still unclear, but athletes can obtain benefits in focus and concentration using slow breathing strategies.

## 1. Introduction

Studies have shown that our breathing techniques can significantly impact sports performance. In humans, it has been documented that our breathing can modulate sympathetic vasoconstrictor activity within the breath [1,2,3,4,5,6]. However, despite the common use of a wide variety of breathing techniques in athletic scenarios, only a few studies have explored the relationship between breathing techniques and sports performance [7,8].

In general, the use of different breathing exercises involving various depths and frequencies is commonly used in different sports. [9,10,11]. However, there remains a large unclear area of how the different depths and frequencies may have a connection with the athlete’s performance, both physiologically and psychologically. Extensive scientific consistency proves that voluntary control of breathing can alter autonomic responses [12,13], and respiratory frequency can differentially affect the function of the cardiovascular system [14]. Slow breathing has been shown to decrease basal heart rates, heart rate responses to standing, and blood pressure. Additionally, it decreases sympathetic activity during altitude-induced hypoxia, improves oxygenation, decreases peripheral chemoreceptor function, and improves exercise performance, baroreflex sensitivity, and baroreflex function [15].

On the other hand, even in the case of fast breathing or high work of breathing, there seems to be a consensus opinion that it can increase alveolar ventilation by 20-fold, resulting in impaired gas exchange and decreased endurance performance. Due to the consequent increase in work, the respiratory system can be limited in functions both by the expiratory flow and by the fatigue of the diaphragm [16]. Moreover, fast breathing in normal individuals can increase blood pressure, heart rate, and the sympathetic drive to the heart [17]. However, some researchers have reported no changes in the cardiovascular parameters or autonomic functions after practicing fast breathing [18].

Although the basic physiology issue provides consistent answers, the question remains of how to determine the optimal depth and frequency of breathing based on the research findings regarding performance. Therefore, the aim of this narrative review is to analyze, in detail, all the possible aspects related to breathing at different frequencies and depths in order to determine whether the hypothesis that it is a valid strategy for achieving optimal performance can be supported.

## 2. Methods

A non-systematic narrative review was carried out with the main objective of summarizing the evidence from the literature, with the purpose of providing a comprehensive overview of sports training for doctors, physiologists, psychologists, technicians, and off-field sports trainers. Our goal was to provide them with this information. The narrative review focuses on the psychological and physiological effects of two types of breathing, different from autonomous and spontaneous breathing: slow voluntary breathing and fast voluntary breathing.

For a better understanding of the two breathing types, we take into account the average respiratory rate in humans, between 10 and 20 breaths per second (0.16 to 0.33 Hz at minimum). Additionally, the result of this effect analysis will be analyzed in relation to the athletes’ performance requirements, and the outcome of this evaluation will then be able to be applied both to training and competition as well as to subsequent recovery periods. There will be a brief introduction to the physiology of normal breathing in each section, followed by a short description of the effects of slow and fast breathing on the human body in a healthy state. It should be noted that slow breathing is defined in this review as 4 to 10 breaths per minute (0.07–0.16 Hz) as a minimum. Additionally, fast breathing is defined as over 20 breaths per minute (0.33 Hz) as a minimum.

First, we searched Medline via PubMed, Scopus, and WoS for articles published in English that addressed our main objectives.

The following search terms were used: “breathing” OR “breathe” OR “lung volume” OR “voluntary respiration” OR “slow breathing” OR “fast breathing” AND “exercise” OR “athlete” OR “performance” OR “sports”. Additional studies were included if they were published in the last 10 years and were referring to the topic of interest. The results obtained were divided into the following two sections, detailing psychological and physiological perspectives:(A)Slow-paced breathing(B)Fast-paced breathing

In addition to the search terms used, we have set a priori the effects of breathing at a rate from 4 to 10 breaths per minute (or 0.07 to 0.16 Hz) as well as at a rate of over 0.33 Hz in humans. Any research that investigated breathing at a rate outside of this range was excluded from our search.

A Medline–Scopus–WoS search was expanded during the writing of the manuscript to include literature relevant to the normal physiology of the respiratory system, as well as the cardiovascular, cardiorespiratory, and autonomic nervous systems, as well as other topics that could be relevant to the revision as it progressed.

## 3. Breathing: Physiological and Mechanical Aspects

As a rule, the functional capacity of a healthy human respiratory system, including the lungs, chest wall, and neural control systems, is greater than the demands placed upon it by heavy exercise [19]. Given the challenges that the respiratory system must overcome during intense exercise, this is an impressive achievement. A healthy respiratory system must regulate alveolar partial pressures of oxygen and carbon dioxide through a constant increase in alveolar ventilation (AV), which is often 20 times greater than resting levels [20]. The physiological cost associated with such an increase in ventilation must be minimized, which is achieved through the respiratory musculature’s capacity for force development [21].

Although the respiratory system is quite capable of overcoming these challenges, it can be limited during exercise in some cases. During exercise, some individuals can suffer from compromised ventilation required for blood gas homeostasis, resulting in high work of breathing. Whenever the lung and chest wall cannot generate adequate flow and volume, expiratory flow limitations (EFL) can occur, resulting in fatigue of the diaphragm. A competitive relationship between the locomotor and respiratory muscles may reduce exercise performance [22,23].

There are two types of work to ventilate the lungs: elastic and non-elastic. An elastic work component involves working against lung elastic recoil, chest wall recoil, and surface tension. As well as airway resistance, tissue resistance contributes to the non-elastic component [24]. Breathing involves inertia forces, gravitational forces, and chest wall distorting forces. Inhalation and expiration result in mechanical work that is elastic and against gravity; flow-resistive work, except for elastic energy stored previously; and negative work [25]. When exercising to exhaustion, minute ventilation (VE) causes a disproportionate increase in breathing work and oxygen consumption. In healthy and fit subjects with normal respiratory function, regression equations demonstrate that the work and oxygen costs of breathing increase in a similar way during progressive-intensity exercise [21].

As emotions change, breathing can also change, such as sadness, happiness, anxiety, or fear. The process of respiration contributes to physiological homeostasis and coexists with emotions [26].

## 4. Breathing Pace and Its Effects on Physiological and Psychological Aspects

The study of respiratory rate modulation and its impact on physiological and psychological effects have been the subject of considerable interest in recent years. The literature reports that slowing down the respiratory rate below 10 breaths per minute, known as slow-paced breathing, and increasing it above 20 breaths per minute, referred to as fast-paced breathing, can elicit distinct physiological and psychological effects.

Numerous studies have demonstrated that slow-paced breathing can reduce sympathetic nervous system activity, decrease blood pressure, and enhance heart rate variability. In contrast, fast-paced breathing can activate the sympathetic nervous system, increase heart rate, and elevate blood pressure.

It is noteworthy that the normal respiratory rate for an average adult is 12 to 20 breaths per minute. Tachypnea, which refers to rapid breathing, can occur in response to various physiological and psychological conditions, including anxiety, fever, and hypoxia [27].

### 4.1. Slow-Paced Breathing

Slow breathing, also known as bradypnea, can have several causes, including medical conditions, medications, chronic obstructive pulmonary disease, heart failure, metabolic disorders, alcohol and drug use, and even sleep [28].

On the other hand, the purpose of voluntary slow-paced breathing (VSB) is to promote physical and mental health, in part through the activation of the vagus nerve, the main nerve of the parasympathetic nervous system [15,29,30].

A key marker of the autonomous nervous system function and a potent predictor of physical morbidity and mortality is heart rate variability (HRV), a measure of the variation in time between each heartbeat. Greater variability indicates a greater ability of the autonomic nervous system to regulate itself. This parameter may be used as a diagnostic and predictive biomarker of mental health since more severe symptoms are significantly associated with reduced HRV [31,32].

HRV findings led to the implementation of a new technique widely used in several physical illnesses and mental disorders: HRV biofeedback (HRVB), a non-invasive therapy training aimed at increasing heart rate oscillations through real-time feedback and slow breathing training [33].

Even though there are various other factors that affect sports performance, we have analyzed the major effects of VSB on physiology and psychology.

#### 4.1.1. VSB: Physiological Perspective

Although it is still debated, investigations into the physiological effects of slow breathing because of voluntary action have uncovered significant effects on the respiratory, cardiovascular, cardiorespiratory, and autonomic nervous systems [15,34].

During VSB, the inhalation and exhalation periods are controlled (“paced”), with exhalation being longer than inhalation. Compared to spontaneous breathing, VSB is usually realized at a pace of around six cycles per minute (cpm), while the spontaneous breathing frequency generally comprises between 12 and 20 cpm with higher amplitude than spontaneous breathing [35]. VSB has been shown to improve health and stress physiology on many levels, including enhancing the function of the autonomic nervous system (e.g., baroreflex, respiratory sinus arrhythmia), cardiopulmonary and neuroendocrine functions, decreasing anxiety and arousal, and increasing resilience. In terms of voluntary slow breathing’s physiological effects, modest reductions in blood pressure were observed following voluntary slow breathing interventions [15].

#### 4.1.2. VSB: Psychological Perspective

Virtual stress reduction training, or VSB, is a process that involves multiple sessions with a professional over a few weeks. The practice is typically supplemented with at-home exercises. The results of VSB have been remarkable, with individuals reporting positive psychological improvements in areas such as stress, anxiety, depression, cognition, fibromyalgia, and reducing substance cravings. Additionally, VSB has been shown to enhance executive functions and improve sports performance, among other benefits [31,32,33,36,37,38,39,40]. A range of training techniques is available, including Eastern traditions, such as meditation and mindfulness, as well as mind-body exercises, such as Tai Chi Chuan (TCC), Yoga, and contemplative activities (ContActs) [30].

### 4.2. Fast-Paced Breathing

Fast breathing is also known as hyperventilation and tachypnea [28]; however, it is important to distinguish these two last conditions. In fact, hyperventilation can be caused by a few factors, including stress and anxiety, respiratory infections, or certain medical conditions such as asthma [41], while tachypnea can be caused by several factors, including fever, physical activity, heart or lung disease, or anemia [27].

As the name suggests, tachypnea is used to describe rapid and shallow breathing, not hyperventilation, which is the process of rapid and deep breathing. The two conditions are similar in that they result from an accumulation of carbon dioxide in the lungs, which in turn leads to a higher level of carbon dioxide in the blood [27]. Fast breathing is characterized by an increased respiratory rate, which is typically defined as more than 12 breaths per minute in adults [42]. In the absence of voluntary action, fast breathing can be caused by a variety of factors, such as cardiac or lung disease, anemia, infections, trauma, neurological conditions, medication side effects, metabolic disorders, and chronic conditions, such as asthma and COPD, COVID-19, or emotional stress [43,44,45].

It is well known in human physiology that when an accumulation of carbon dioxide is present in the blood, it causes the blood to become acidic, causing the brain to be alerted. Therefore, in response to the imbalance in the blood pH, the brain alerts the respiratory drive to increase in pace to correct it. By doing so, the blood pH can return to within the normal range in acidity again.

As well as VSB, even voluntary fast-paced breathing (VFB) has effects on both physiology and psychology.

#### 4.2.1. VFB: Physiological Perspective

VFB, through the activation of the sympathetic nervous system, can have several negative physiological effects on the body. It can lead to decreased levels of carbon dioxide in the blood, which can cause several symptoms, such as lightheadedness, dizziness, tingling in the fingers and toes, and shortness of breath. Fast breathing can also cause the blood vessels to constrict, which can lead to decreased blood flow to the brain and other vital organs [21], causing feelings of anxiety and panic. Fast breathing can also cause the body to increase carbon dioxide, which can lead to an alkalosis state; this state can lead to muscle cramps and spasms related to the nervous system [46,47].

Fast breathing can lead to decreased oxygen levels in the body, which can cause several symptoms, such as shortness of breath, fatigue, and confusion [48]. In severe cases, it can lead to hypoxia, which is a condition in which the body does not have enough oxygen to function properly [49]. Fast breathing can also cause the body to release stress hormones, which can lead to feelings of anxiety and panic. In some cases, fast breathing may cause chest pain and other symptoms that require medical attention [50].

#### 4.2.2. VFB: Psychological Perspective

Fast breathing can have a significant impact on an individual’s mental state and well-being because a decrease in the level of carbon dioxide in the blood, FB, and also VFB can cause feelings of anxiety and panic [51]. This condition can trigger a “fight or flight” response in the body, releasing stress hormones such as adrenaline, which can further increase the heart and breathing rates [52].

From a physio-psychological perspective, fast breathing can also lead to decreased levels of oxygen in the body, which can cause feelings of lightheadedness, dizziness, and confusion. In some cases, it can lead to feelings of claustrophobia and a sensation of not being able to breathe [51].

Additionally, fast breathing can be a physical manifestation of stress and anxiety, which can then further contribute to feelings of stress and anxiety. In some cases, fast breathing may also be a symptom of a panic attack, which can cause severe distress and affect an individual’s quality of life [53].

## 5. Perspective of the Athlete on Breathing and Practical Applications in Sports

VSB, through the activation of the parasympathetic nervous system, can have a positive impact on an athlete’s performance. In terms of purely athletic performance, physiological and psychological factors must be considered in relation to an athlete’s competition rather than considering their well-being and mental health. Slow breathing can help lower the heart rate and blood pressure, which can improve cardiovascular fitness and endurance. Additionally, VSB can also help to reduce stress and anxiety, which can improve an athlete’s mental state and ability to perform under pressure. Slow breathing can also help to increase the amount of oxygen that reaches the body’s cells, which can improve overall health and well-being. Additionally, slow breathing can help to improve the function of the respiratory system, which can help to reduce the symptoms of certain respiratory conditions, such as asthma and chronic obstructive pulmonary disease (COPD), that might affect the athlete’s performance. Diaphragmatic breathing can also improve the athlete’s lung capacity and the efficiency of their breathing, which can help to improve their endurance and stamina, allowing them to perform better for a longer time period. Over the long term, VSB can enhance the ability to regulate emotions and increase resilience to stress [34]. Heart rate variability increases as well, and this can result in better adaptability to the environment [15,54]. Finally, VSB can also improve executive functions in athletes with a better impact on attention, working memory, and cognitive flexibility [55].

On the other hand, fast breathing, or VFB, as hyperventilation, has negative effects on performance and has been a subject investigated since the last century [56]. Hyperventilation, in fact, can cause the body to lose too much carbon dioxide, which can lead to a condition called respiratory alkalosis [57]. This can cause a decrease in the amount of oxygen that reaches the muscles, which can lead to fatigue and decreased endurance. Hyperventilation can also cause the blood vessels to constrict, which can reduce blood flow to the brain and other vital organs. This can lead to decreased coordination and balance, which can affect an athlete’s ability to perform complex movements. Additionally, hyperventilation can cause feelings of anxiety and panic, which can negatively impact an athlete’s mental state and ability to perform under pressure [58]. At the same time, controlled breathing techniques, such as fast-paced breathing, can help to improve an athlete’s performance. VFB can increase oxygen delivery to muscles [42] and can improve strength, speed, reaction time [45,59], and abdominal muscle tone [60].

### 5.1. Breathing in Sport: Examples of Practical Applications

In the following lines, some practical applications are reported for breathing pace management in sports based on the previous considerations.

#### 5.1.1. Before Exercise

Before exercise, it is recommended to practice breathing suitable for the performance that the athlete will have to sustain.

Slow, deep breathing before exercise can have several benefits if the following conditions are needed: (1) increased oxygenation; (2) relaxation; (3) improved focus; (4) better posture. If increased oxygenation is needed, slow and deep breathing can help increase the level of oxygen in the body, which can improve athletic performance. If relaxation is needed, deep breathing can help calm the nervous system, reducing feelings of stress and anxiety before exercise. If improved focus is needed, practicing slow and deep breathing can help improve focus and concentration, allowing for better athletic performance. Finally, if better posture is needed, diaphragmatic breathing helps promote good posture, which can improve athletic performance by reducing tension and improving alignment.

For practicing slow and deep breathing before exercise, taking slow and deep breaths through the nose, filling the diaphragm with air, and slowly exhaling through the mouth is recommended. This can be achieved while standing, sitting, or lying down and can be performed for a few minutes before starting the exercise.

In summary, slow, deep breathing, also known as diaphragmatic breathing or abdominal breathing, is recommended before exercise as it can help increase the oxygen level in the body, promote relaxation, improve concentration, and improve posture.

Fast breathing before exercise can have several benefits if the following conditions are needed: (1) greater activation of the sympathetic nervous system; (2) higher heart rate; (3) greater activation of the athlete with a controlled effect of the “fight or flight” phenomenon.

In summary, if the athlete has to face performances where it is necessary to create greater physiological and psychological activation, encouraging rapid breathing may be more appropriate. 

#### 5.1.2. During Exercise

It is important to maintain a steady and controlled breathing pattern. This helps regulate the amount of oxygen and carbon dioxide in the body, ensuring adequate oxygen supply to the muscles and the proper elimination of waste products.

Two common types of breathing during exercise are (1) rhythmic breathing and (2) controlled breathing. Rhythmic breathing involves inhaling and exhaling in a controlled, rhythmic pattern, typically in time with the movement of the exercise. For example, when running, rhythmic breathing might involve inhaling for three steps and exhaling for two. Controlled breathing involves inhaling deep into the diaphragm, then exhaling slowly and completely. Controlled breathing can help regulate the heart rate and reduce feelings of stress and anxiety during exercise.

It is important to find a breathing pattern that works for an individual and follow it throughout the exercise. It is also important to listen to your body and adjust your breathing as needed to ensure comfort and proper oxygen delivery to your muscles.

In summary, while exercising, it is important to maintain a steady and controlled breathing pattern, such as rhythmic breathing or controlled breathing, to regulate the amount of oxygen and carbon dioxide in the body and to ensure proper oxygen delivery to the muscles.

#### 5.1.3. After Exercise

After exercising, it is important to practice slow, deep breathing to help your body recover and return to a calm state.

Slow, deep breathing after exercise can have several benefits, including a reduced heart rate, better recovery, relaxation, and improved posture.

In fact, slow and deep breathing can help slow the heart rate, thus reducing stress on the cardiovascular system, as well as helping to improve the body’s ability to recover from exercise by facilitating the removal of waste products, such as lactic acid, from muscles. Moreover, deep breathing can help calm the nervous system, thus reducing feelings of stress and anxiety after exercise. Finally, diaphragmatic breathing helps promote good posture, which can improve recovery by reducing tension and improving alignment.

## 6. Discussion

The connection between respiratory rate and sports performance is complex and multifaceted, although a lack of specific data must be underlined. The main aim of this narrative review was to highlight the possible effects of the main breathing strategies, merging the physical, physiological, and mental effects.

In high-level and Olympic sports, the physio-psychological conditions necessary for the athletes are different; therefore, breathing must be controlled at the correct frequency based on the results we expect to aim for the best performance.

Indeed, the respiratory rate during exercise can affect athletic performance in several ways:Oxygen delivery: the respiratory rate determines the amount of oxygen that is delivered to the muscles during exercise, which is critical for peak athletic performance.Carbon dioxide elimination: the respiratory rate also determines the amount of carbon dioxide that is eliminated from the body during exercise, which can help improve athletic performance.Heart rate regulation: the respiratory rate can also affect the heart rate, which is important for peak athletic performance and reducing feelings of stress and anxiety during exercise.Concentration and focus: the breathing rate can also affect the athlete’s focus and concentration during exercise. Controlled, rhythmic breathing can help improve focus and concentration, allowing for better athletic performance.

It is important to note that the optimal breathing rate during exercise can vary according to the individual and the type of activity or sport. In general, it is recommended to find a breathing rate that is comfortable and which allows for adequate oxygen delivery to the muscles while maintaining proper waste product elimination.

The respiratory frequency takes on two diametrically opposite characteristics in terms of optimal performance. Slow breathing can benefit athletes in a variety of ways, not just physically but mentally as well. It can help improve cardiovascular fitness, reduce stress and anxiety, and improve overall health and well-being. It can help athletes maintain focus and concentration during training and competition. Rapid breathing can have a significant impact on an individual’s mental state and well-being. It can cause feelings of anxiety, panic, dizziness, and lightheadedness and trigger a stress response in the body. If it is not voluntary, it is important to consult a doctor if it is accompanied by other symptoms or affects the athlete’s quality of life.

### Limitations

Although a detailed description of the breathing strategies is provided, some limitations of this paper must be underlined. This narrative review cannot definitively answer the question about the relationship between the type of breathing and sports performance. In fact, there were some limitations between the studies in terms of sample size, duration of the intervention, presence or absence of breath professionals, consistent evidence of the benefit of breathing on the performance of athletes, and consistent evidence that slow breathing had both positive physiological and psychological effects. Moreover, it is unclear how fast breathing impacts athletes and whether it can be offered to them during training or competitions due to large limitations on the effects of fast breathing. The limitations of this narrative review could be transformed into investigation areas for the future.

## 7. Conclusions

In summary, the role of breathing in the athlete’s performance cannot be underrated, but at the same time, a paucity of scientific data is still present to draw a clear and definitive picture of the phenomenon. Athletes can be stimulated to practice controlled breathing throughout training sessions and during competitions in order to optimize their performance, specifically, to improve their mental and physical well-being, maintain focus and concentration, achieve their goals, and maintain a positive attitude.

Despite the complexity of the relationship between respiratory rates and sports performance, maintaining a consistent and controlled respiratory rate during exercise can assist with ensuring adequate oxygen supply to the muscles, proper elimination of waste products, regulation of the heart rate, and improved focus and concentration, all of which are essential for achieving peak athletic performance.

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
