# Peer review of "Sports Performance and Breathing Rate: What Is the Connection? A Narrative Review on Breathing Strategies"

_sports, 2023, doi:10.3390/sports11050103_

Round 1

Reviewer 1 Report

Sports performance and breathing rate: what is the connection?

The abstract presents the purpose of the review article, but it doesn’t describe what was reviewed.

-How many studies were selected for this review?

-What was the main outcome of the review?

This is a good review of the physiological and mechanical aspects of breathing as well as a good explanation of fast and slow-paced breathing.

Furthermore, the authors describe the practical application of breathing in sports.

This review doesn’t describe how the studies were selected or any of the methodology parts. It is more like a literature review which is important for athletes and coaches, but it doesn’t follow the traditional guidelines for a review.

Author Response

We would like to thank you for the time allowed to this review process. As a result, we are submitting the revised version for a possible publication in this respectable Journal. Below, you can find our responses; each comment is followed by its respective reply. We made changes in the manuscript in order to address suggestions and make it clearer for the readers, we underlined in yellow and pink the parts where more text was added to increase the word number, while we used the track changes for the responses to your comments. All authors have made sufficient contributions and have approved the submitted manuscript.

Best regards,

The Authors

Legend:

R1(Reviewer 1)

A (Authors)

1) R1:

The abstract presents the purpose of the review article, but it doesn’t describe what was reviewed.

A:

We modified the abstract to be clearer to avoid any misleading concept. Lines 13-16.

2) R1:

How many studies were selected for this review?  

A:

The review is a narrative one, in fact we corrected the title because we recognize it was not clear. Anyway, a better description of the literature searching strategies was included in “2. Methods” section.

3) R1:

What was the main outcome of the review?

A:

The main outcome is a new and integrated point of view on the relationship between breathing and sport performance.

4) R1:

This is a good review of the physiological and mechanical aspects of breathing as well as a good explanation of fast and slow-paced breathing.

Furthermore, the authors describe the practical application of breathing in sports.

A:

Thank you, we are glad that our point of view could be positively considered.

5) R1:

This review doesn’t describe how the studies were selected or any of the methodology parts. It is more like a literature review which is important for athletes and coaches, but it doesn’t follow the traditional guidelines for a review.

A:

The paper is a narrative review anyway a Methods section was improved, describing the methodology used related to the inclusion/exclusion studies.

Reviewer 2 Report

As per notes attached

Author Response

We would like to thank you for the time allowed to this review process. As a result, we are submitting the revised version for a possible publication in this respectable Journal. Below, you can find our responses; each comment is followed by its respective reply. We made changes in the manuscript in order to address suggestions and make it clearer for the readers, we underlined in yellow and pink the parts where more text was added to increase the word number, while we used the track changes for the responses to your comments. All authors have made sufficient contributions and have approved the submitted manuscript.

Best regards,

The Authors

Legend:

R2 (Reviewer 2)

A (Authors)

1) R2:

Title

It is too brief and does not describe what was done in the study. It suggests that the title have what was evaluated, and even that we can identify the dependent and independent variable.

A:

The title was modified adding “a narrative review on breathing strategies”, which is our main goal. Thanks for this comment useful to avoid any misleading consideration.

2) R2:

Abstract

It is very superficial, does not comply with the journal's norms and still does not present the main results found. It would be feasible to comply with the journal's norms, where we could identify the Background, objectives, methodology, results and conclusions, including practical applications of the findings.

I ask you to confirm that the keywords are found as descriptors in health sciences.  

A:

The abstract resume the content of the narrative review. According to the guidelines for Authors, it has a background containing the aim (Lines 10-16), no methods and results are mentioned because it is a narrative review, therefore the abstract continues with a detailed description of the content of the review. A brief and practical conclusion has been added, thanks for this suggestion.

Finally, we confirm that the keywords can be considered as descriptors in health sciences.   

3) R2:

Introduction

The study is characterized as a review. However, in the introduction we cannot evaluate the proposal. That is, the introduction should be written from general to specific, where around three to five paragraphs the authors would describe from general to specific what they intend to study and review.

The first paragraph would be a contextualization of what is intended to be studied, covering more general topics of the subject.

The second or the second and third paragraphs would be the support or scientific pillars of what was intended, making the connection from the general to the specific.

The third or fourth and fifth paragraph would be destined to show the problem. This problem must be presented clearly, presenting the studies for and against what is intended. Mentioning that there are few studies is not an acceptable scientific shortcoming.

From the gap, the objectives of the study would be presented and the hypotheses to be answered by the research would be presented.

A:

Thanks for your suggestion. According to you comment we modified the introduction section as you stated and we wrote better the aim of the paper in the final part of the introduction section.

4) R2:

Methods

It should more clearly present the design of the study. A CONSORT or strobe, should be presented in order to get a better view of the study design.

The research criteria are not presented in the study, such as keywords, researched database, research language, period, among others.

The guidelines used and the quality standards used were not mentioned. How was the scientific quality of the bibliography consulted determined?

Was it not mentioned how the main studies that were presented were identified, much less how they were chosen?

The type of review study proposed was not mentioned.

The study does not have a topic described as methodology. How can we replicate the study?

Results

The results were not presented. Much less were the choices based on technical standards described in the methodology.

The main studies that would be presented in the results were not mentioned, much less how they were chosen. This already mentioned the methodology. The study simply lacks results.

A:

We understand the comment, at the same time our paper is a narrative review about the topic, this is the reason why no methods (CONSORT or strobe) are described. A Methods section was added, and we described with details our procedures. The results are substituted by the “Relevant sections” as stated in the Instructions for Authors “The structure can include an Abstract, Keywords, Introduction, Relevant Sections, Discussion, Conclusions, and Future Directions”. We used a very similar style respect previous papers published on MDPI (https://www.mdpi.com/2624-5175/5/2/20; https://www.mdpi.com/2075-4663/11/2/38).

5) R2:

Discussion

An item called discussion was not presented, making it difficult to determine what will be effectively important in terms of research for the study. Limitations were not mentioned or even the main intended outcomes were explained.

A:

A Discussion section and a “Limitations” section have been included. Thanks for this suggestion, will be very useful for the readers.

6) R2:

Conclusion

Are presented, however without basis for this. This reflects the study, which, considering that it did not have a logical sequence, ends up being weakened in the conclusions.

A:

The conclusion section has been modified to better clarify it.

7) R2:

References

Please confirm the formatting of the references and the 60 references presented, 23 are current and 37 have more than five years of publication. Please update the theoretical framework, in view of the topic update. There are many books in the references which ends up making the work unfeasible.

A:

The formatting is correct. We understand you comment about the books, but we want to underline the paucity of information on this topic, for this reason we decided to submit a narrative review, in order to stimulate qualitative experimental research on this important and interesting topic. Moreover, as stated in the Methods section, we also used some other kind of resources “Classic papers and other relevant papers and books were also included if they were cited in studies included in the present review.”

8) R2:

Overview

The manuscript addresses presented a relevant research topic.

It would be advisable to do a general review. The work needs all reviewed and redone, to consider its publication.

A:

Thanks for understanding the importance of the topic. Our paper is a narrative review, therefore according to your comment we made changes in the main document. We hope that you would judge our changes in the paper in a positive way.

Reviewer 3 Report

Dear all,

The manuscript fits with the aim of the Sports Journal, and the subject reveals good content for researchers and professionals. However, some points are listed below:

Title

No comment.

Abstract

The Abstract didn’t explain how the study was done, didn’t include model organisms used, or methodological details. Authors used abbreviations (e.g., VSB and VFB) without presenting the abbreviated terms.

1. Introduction

The authors didn’t follow the structure of the journal, for instance, I couldn’t realize where the introduction section ended, and move to the next section. Also, the introduction didn’t fill the gap, furthermore, it doesn’t conclude purposes and/or hypothesis.

2. Materials and Methods

This section is missing.

The experimental approach to the problem should be presented.

As the paper is review article, a flow chart of the search strategy and retrieval of articles should be added, containing the identification, screening, eligibility, and including and excluding criteria for the reviewed papers.

Statistical analyses should be displayed.

3. Results

This section is missing.

Study Selection, characteristics of included studies, levels of evidence, methodological quality, participants, and interventions should be shown.

This point: 3.1. Breathing in sport: examples of practical applications; it needs to be summarized, and presenting it in bullets isn’t appropriate for the structural design of the journal.

4. Discussion

This section is missing.

5. Conclusions

Please eliminate redundancy. State the conclusion with a brief description of the paper's purpose, major findings with an explanation, and recommendations for prospective studies.  

References

No comments.

Regards, 

Author Response

We would like to thank you for the time allowed to this review process. As a result, we are submitting the revised version for a possible publication in this respectable Journal. Below, you can find our responses; each comment is followed by its respective reply. We made changes in the manuscript in order to address suggestions and make it clearer for the readers, we underlined in yellow and pink the parts where more text was added to increase the word number, while we used the track changes for the responses to your comments. All authors have made sufficient contributions and have approved the submitted manuscript.

Best regards,

The Authors

Legend:

R3 (Reviewer 3)

A (Authors)

1) R3:

The Abstract didn’t explain how the study was done, didn’t include model organisms used, or methodological details. Authors used abbreviations (e.g., VSB and VFB) without presenting the abbreviated terms.

A:

The paper is a narrative review for this reason the abstract did not explain any specific methodology. About the abbreviations the terms are presented in lines 16-17, we modified the sentence to be clearer.

2) R3:

The authors didn’t follow the structure of the journal, for instance, I couldn’t realize where the introduction section ended, and move to the next section. Also, the introduction didn’t fill the gap, furthermore, it doesn’t conclude purposes and/or hypothesis.  

A:

According to the guidelines for Authors we divided the sections of the paper as requested: “The structure can include an Abstract, Keywords, Introduction, Relevant Sections, Discussion, Conclusions, and Future Directions”. Following your suggestions, we modified the sequence as: Introduction is the point 1; then we added a Methods section, point 2; then we have the Relevant sections, points 3-5; we add Discussion point 6 and Conclusions is the point 7. 

3) R3:

  1. Materials and Methods

This section is missing.

The experimental approach to the problem should be presented.

As the paper is review article, a flow chart of the search strategy and retrieval of articles should be added, containing the identification, screening, eligibility, and including and excluding criteria for the reviewed papers.

Statistical analyses should be displayed.

  1. Results

This section is missing.

Study Selection, characteristics of included studies, levels of evidence, methodological quality, participants, and interventions should be shown.

This point: 3.1. Breathing in sport: examples of practical applications; it needs to be summarized, and presenting it in bullets isn’t appropriate for the structural design of the journal.

A:

Thanks for the comment. We understand you point and we add a Method section “2. Methods” where we explaine our methodology for the narrative review and where we explicit how we organized the Relevant sections of the review. We used a very similar style respect previous papers published on MDPI (https://www.mdpi.com/2624-5175/5/2/20; https://www.mdpi.com/2075-4663/11/2/38).

According to this choice, the “Results” section is not present as name but the Relevant sections (points 3-5) could be considered the same.

4) R3:

  1. Discussion

This section is missing.

A:

We modified the sequence of the sections and we add discussion.

5) R3:

Please eliminate redundancy. State the conclusion with a brief description of the paper's purpose, major findings with an explanation, and recommendations for prospective studies.

A:

Done.

Round 2

Reviewer 1 Report

I am happy with the changes. I have no further concerns or suggestions. 

Author Response

Legend:

R1(Reviewer 1)

A (Authors)

1) R1:

I am happy with the changes. I have no further concerns or suggestions.

A:

We are grateful for the time spent in the review process and we are glad you appreciated the actual version of the paper.

Reviewer 2 Report

We thank the authors for the adjustments made, which greatly improved the presentation of the manuscript. I only request that the main absolute and statistical values be inserted in the abstract. At most I consider the manuscript in conditions to be published.

Author Response

Legend:

R2 (Reviewer 2)

A (Authors)

1) R2:

We thank the authors for the adjustments made, which greatly improved the presentation of the manuscript. I only request that the main absolute and statistical values be inserted in the abstract. At most I consider the manuscript in conditions to be published.

A:

We are very grateful for your suggestions and comments that were fundamental to improve the quality of the paper. We are glad you appreciated our work on the paper.  About your last request, we do not have statistical values to insert in the abstract because the paper is a Narrative review and not a Systematic one.

Reviewer 3 Report

Dear all,

The manuscript fits with the aim of the Sports Journal, and the subject reveals good content for researchers and professionals.

Thanks for doing the amendments.

But Results section still missed. Study Selection, characteristics of included studies ‘inclusion criteria’, levels of evidence, methodological quality, participants, and interventions hasn't showed yet.

Best regards,

Author Response

Legend:

R3 (Reviewer 3)

A (Authors)

1) R3:

The manuscript fits with the aim of the Sports Journal, and the subject reveals good content for researchers and professionals.

Thanks for doing the amendments.

But Results section still missed. Study Selection, characteristics of included studies ‘inclusion criteria’, levels of evidence, methodological quality, participants, and interventions hasn't showed yet.

A:

We are very grateful for your suggestions and comments that were fundamental to improve the quality of our paper. The actual version of the Narrative review is more and more qualitative respect the original one. We think this is the essence of the review process. Regarding the last comment:

  • Because the paper is a Narrative Review the “Results Section” is not required according to the Instructions for Authors (https://www.mdpi.com/journal/sports/instructions), we send two examples of other papers published in MDPI as Narrative Reviews https://www.mdpi.com/2624-5175/5/2/20, https://www.mdpi.com/2075-4663/11/2/38
  • “Study Selection, characteristics of included studies ‘inclusion criteria’” are described in “Methods Section” Lines: 64-105
  • “levels of evidence, methodological quality, participants, and interventions” we understand the comment and we would agree is the paper was a systematic review but this is a Narrative Review and this last part is not usually required or present in other Narrative Reviews.